Collaborative filtering based on GNN with attribute fusion and broad attention

Liu MingXue 1
Wang Min 1 2 wangmin@gnnu.edu.cn
Li Baolei 1 2
Zhong Qi 1
1 School of Mathematics and Computer Science, Gannan Normal University , Ganzhou , China
2 Key Laboratory of Data Science and Artificial Intelligence of Jiangxi Education Institutes, Gannan Normal University , Ganzhou , China
Angiulli Giovanni
Electronic publication date: 2025 Feb 25
Publication date: 2025
Volume: 11
Electronic Location ID: e2706
Received 2024 Nov 20; Accepted 2025 Jan 24
Copyright: © 2025 Liu et al.
Copyright year: 2025
Copyright holder: Liu et al.
License: This is an open access article distributed under the terms of the Creative Commons Attribution License, which permits unrestricted use, distribution, reproduction and adaptation in any medium and for any purpose provided that it is properly attributed. For attribution, the original author(s), title, publication source (PeerJ Computer Science) and either DOI or URL of the article must be cited.
License URL: https://creativecommons.org/licenses/by/4.0/

Keywords: Graph neural networks, Collaborative filtering, Attribute fusion, Broad attention, Cross interaction

Funding: National Natural Science Foundation of China 62362002 Natural Science Foundation of Jiangxi Province 20224BAB212022 This work was supported by the National Natural Science Foundation of China (No. 62362002) and the Natural Science Foundation of Jiangxi Province (No. 20224BAB212022). The funders had no role in study design, data collection and analysis, decision to publish, or preparation of the manuscript.

==============================
Recommender systems based on collaborative filtering (CF) have been a prominent area of research. In recent years, graph neural networks (GNN) based CF models have effectively addressed the limitations of nonlinearity and higher-order feature interactions in traditional recommendation methods, such as matrix decomposition-based methods and factorization machine approaches, achieving excellent recommendation performance. However, existing GNN-based CF models still have two problems that affect performance improvement. First, although distinguishing between inner interaction and cross interaction, these models still aggregate all attributes indiscriminately. Second, the models do not exploit higher-order interaction information. To address the problems above, this article proposes a collaborative filtering method based on GNN with attribute fusion and broad attention, named GNN-A2, which incorporates an inner interaction module with self-attention, a cross interaction module with attribute fusion, and a broad attentive cross module. In summary, GNN-A2 model performs inner interactions and cross interactions in different ways, then extracts their higher-order interaction information for prediction. We conduct extensive experiments on three benchmark datasets, i.e., MovieLens 1M, Book-crossing, and Taobao. The experimental results demonstrate that our proposed GNN-A2 model achieves comparable performance on area under the curve (AUC) metric. Notably, GNN-A2 achieves the optimal performance on Normalized Discounted Cumulative Gain at rank 10 (NDCG@10) over three datasets, with values of 0.9506, 0.9137, and 0.1526, corresponding to respective improvements of 0.68%, 1.57%, and 2.14% compared to the state-of-the-art (SOTA) models. The source code and evaluation datasets are available at: https://github.com/LMXue7/GNN-A2.

Introduction

Recommender systems help manage information overload by filtering data, and analyzing user behavior to improve experiences and drive business growth (Zhu et al., 2024). They are widely used in social networks (e.g., Facebook), short video platforms (e.g., TikTok), and e-commerce (e.g., Amazon, Taobao). Collaborative filtering (CF), a core algorithm in recommender systems, uses historical interactions to learn user and item representations, predicting future user behaviors. CF-based recommender systems have been a key research focus (Pan et al., 2020; Luo et al., 2020).

Early matrix decomposition-based recommender systems, as demonstrated by Koren, Bell & Volinsky (2009), He et al. (2017), have developed embeddings by analyzing user-item interaction data, including browsing behavior and rating, to establish user-user and item-item similarities. However, the performance of matrix decomposition methods is limited when interactions are sparse (Cheng et al., 2018, 2016). Therefore, some CF methods try to improve prediction by introducing user and item attribute information, such as the user’s age and the item’s price, which helps to capture the user’s interests and the item’s features more accurately. Afterward, the CF method further adopts attribute embedding techniques to more accurately capture the subtle collaboration information between users and items by analyzing the attribute co-occurrence patterns. These methods have significantly improved the accuracy of prediction (Su, Erfani & Zhang, 2019). Attribute embedding techniques have thus been shown to significantly improve the accuracy of prediction results (Rendle, 2010; Song et al., 2019). For example, in the context of shopping platforms, women aged 25 to 35 years are the primary consumers of skincare products. In this scenario, the attribute interaction <25–35 years old, female> is likely to be more beneficial to prediction than when these two attributes are considered independently.

Factorization Machine (FM), as introduced by Rendle (2010), is widely regarded as a standard approach for addressing multi-field sparse data in both academic and industrial contexts. Subsequently, numerous enhanced versions of FM appeared. Field-aware FM (FFM) (Juan et al., 2016) is one of them, which considers feature interactions between different fields to achieve better prediction results. However, the neglect of nonlinearities and higher-order feature interactions limits the predictive accuracy of these models. Therefore, many deep learning-based models have been proposed to address nonlinear attribute relationships and higher-order feature interactions. Notable models include Neural Factorization Machine (NFM) proposed by He & Chua (2017), Deep & Cross introduced by Geyik, Ambler & Kenthapadi (2019), DeepFM developed by Guo et al. (2017), xDeepFM presented by Lian et al. (2018), Automatic Feature Interaction (AutoInt) as described by Song et al. (2019), and Feature Importance and Bilinear feature Interaction NETwork (FiBiNET) (Huang, Zhang & Zhang, 2019). All of these models attempt to analyze and optimize complex feature interactions. Although these models significantly improve prediction accuracy, uniform feature processing or dependence on specific modules to identify important features leads to a lack of transparency and explanatory power in their methods. In addition, certain useless feature interactions may affect the stability of the model.

Recently, the rise of graph neural networks (GNNs), presented by Li et al. (2015), has provided new tools for dealing with higher-order feature interactions. Feature interaction GNN (Fi-GNN) (Li et al., 2019) translates feature interaction modeling into interaction processes between nodes in a graph structure. Su et al. (2021a) further employed GNN to effectively capture complex feature interaction information and utilized graph learning techniques for feature aggregation. Graph Matching-Based Collaborative Filtering (GMCF) (Su et al., 2021b) distinguishes between inner interactions (user or item attributes alone) and cross interactions (between user and item attributes). However, GMCF still aggregates all attributes indiscriminately, which lacks sufficient explanation. For example, in the scenario of predicting the probability that young women prefer whitening skin care products, the inner interaction <female, 25–35 years old> is definitely more important than that of <female, teacher>. Additionally, the cross interaction <25–35 years old, whitening> is also more important than <25–35 years old, $100–300>. Simple aggregation strategies may weaken the expressiveness of attribute interactions, and fail to fully understand the impact of aggregation of neighboring nodes on the nodes themselves. Moreover, GMCF overlooks higher-order interaction information, which is crucial for improving recommendation performance by leveraging attribute dependencies.

To address the aforementioned problems, this article proposes a CF method based on GNN with attribute fusion and broad attention, named GNN-A2. This approach models inner interactions and cross interactions within graph structures in different ways. It then aggregates these interactions separately. Finally, it extracts higher-order interaction information for prediction. Specifically, we first model the inner interactions through a GNN-based message passing mechanism in the user graph and the item graph, respectively. To capture the varying impacts of different attributes on feature learning, we employ a self-attention mechanism to weight the aggregation of these inner interaction attributes accordingly. At the same time, we propose the attribute fusion strategy to model cross interaction, which assesses the correlation between user attributes and item attributes by cosine similarity calculation, thereby assigning weights to cross interaction attributes, and then performing weighted aggregation. This highlights the varying significance of different attributes, thereby enhancing the matching results of the recommender system. In addition, GNN-A2 fuses information from inner interactions and cross interactions by introducing a broad attentive cross module. The module dynamically evaluates the degree to which different feature interactions contribute to the final prediction and then assigns the appropriate weights. Thus, GNN-A2 can capture meaningful higher-order interactions between users and items, rather than simply treating them as independent factors. This helps to better reveal more complex dependencies among features, thereby enhancing the model’s predictive accuracy and generalizability. To validate the effectiveness of GNN-A2, we performed extensive experiments on three real-world datasets, including MovieLens 1M, Book-crossing, and Taobao. The experimental results show that GNN-A2 demonstrates superior performance compared to several baseline models.

Our main contributions can be summarized as follows. We design an attribute fusion strategy for cross interaction, while fusing inner interactions based on the self-attention mechanism. Furthermore, we optimize the prediction accuracy of the recommender system by assigning appropriate weights to attributes for weighted aggregation.

A broad attentive cross module is introduced to optimize higher-order feature interactions, dynamically learning attribute weights to efficiently enhance the model’s predictive accuracy and generalization capability.

Extensive experiments on three public datasets show that our model outperforms others, highlighting the contributions of the self-attention mechanism, attribute fusion, and broad attentive cross module.

Related work

In this section, we dive deeper into related work on recommendation systems based on GNNs, recommendations based on attribute information, and broad learning mechanisms.

Recommendation systems based on GNN

The graph consists of nodes and edges, the edges representing connections between nodes. In the past decade, graph representation learning, acquisition, and application have become a hot research topic (Ji et al., 2022). GNNs greatly facilitate the learning of entities and their relationships (Kipf & Welling, 2016; Pang, Zhao & Li, 2021; Zhao et al., 2021) by performing neural network operations on the graph structure to obtain a representation of the nodes.

Existing research has employed GNN across a diverse range of tasks, including action recognition (Shi et al., 2019), semantic segmentation (Wang et al., 2019a), image recognition (Chen et al., 2019), recommender system (Pan et al., 2022), and so on. In recent years, GNN has demonstrated significant potential in CF by effectively modeling interactions between users and items. Some works consider user-item interactions as bipartite graphs, utilizing edges between user nodes and item nodes to represent interactions, e.g., buying and ratings (Wang et al., 2019c; Berg, Kipf & Welling, 2017). The focus of these models is the analysis of user-item interactions in GNNs. Other research employs GNNs to model knowledge graphs for the purpose of enhancing recommendation systems (Wang et al., 2019b; Xian et al., 2019). These models treat edges between attributes and users (items) as predefined relationships, rather than relationships between attributes. Nevertheless, existing GNN-based models encounter several challenges, including the sparsity of user-item interaction graphs and their limited capacity to adequately capture higher-order interaction information. To address these issues, recent research has introduced new approaches, such as self-supervised hypergraph transformer (SHT) (He et al., 2023), achieves significant results in handling sparse and noisy data by introducing hypergraph transformers and self-supervised learning components, and candidate-aware graph contrastive learning (CGCL) (Xia, Huang & Zhang, 2022) improves embedding quality by constructing more semantically meaningful contrast pairs. Our approach, in contrast, advances further by incorporating a GNN-based self-attention mechanism to enhance the optimization of attribute interactions. This hybrid approach enables our model to enhance embedding quality by taking attribute interactions into account, while also allowing for the selective emphasis on significant attributes. This results in a more robust and comprehensive representation of the relationships between users and items.

Recommendations based on attribute information

The study of primitive attributes reveals the intrinsic features of both users and items, as well as the behavioral tendencies exhibited by users (Xu et al., 2023; Su et al., 2022). To improve the potential feature representation of users and items, some researchers have integrated attribute information into GNNs (Wu et al., 2020; Zheng, Li & Liao, 2021).

Zheng, Li & Liao (2021) developed a heterogeneous type-specific representation learning approach, which integrates attributes and interaction behaviors, emphasizing the semantic embedding of both users and items. However, given the sparsity of user behavioral data, these approaches based on attribute embedding tend to focus more on graph structure than attribute interactions in practical applications. To explore attribute interactions, Li et al. (2019) and Su et al. (2021a) utilize GNNs as a graph learning process to model and aggregate attribute interactions. However, they treat all attribute interactions as consistent operations and do not effectively utilize the attribute interaction structure information, thus failing to achieve good predictions through joint decision-making. Subsequently, GMCF proposed by Su et al. (2021b) improves the attribute interaction model by separately modeling inner and cross interactions. However, it treats attribute interactions in both inner and cross interactions with equal consideration and aggregates them indiscriminately. This simple aggregation limits the expressive capacity of attribute interactions, and fails to learn the effects of aggregation of neighboring nodes on the nodes themselves. Additionally, GMCF does not take full advantage of higher-order interaction information. GNN-A2 employs a self-attention mechanism to facilitate the fusion of inner interaction attributes. In addition, GNN-A2 incorporates an attribute fusion strategy that assigns appropriate weights to the cross attributes, enabling weighted aggregation based on the degree of correlation between user attributes and item attributes.

Broad learning mechanism

Most deep learning networks require a long training cycle due to their complex deep structure and numerous connection weights. To solve this problem, Pao & Takefuji (1992) designed a random vector function-link neural network (RVFLNN). RVFLNN extends the conventional single-layer feedforward neural network by transforming hidden nodes into enhancement nodes. It establishes connections between the output node and both the input nodes and the enhancement nodes. In addition, random generation is used to obtain the transformation parameters from the input nodes to the enhancement nodes, leading to a substantial enhancement in the performance of the network. To solve the problem of the long training process and high-dimensional data, inspired by RVFLNN and the dynamic stepwise updating algorithm (Chen & Wan, 1999), broad learning system (BLS) (Chen & Liu, 2018) is proposed as an alternative to the deep structure.

The BLS model employs a flat network architecture, wherein the original feature nodes are redefined as mapping features. The network architecture is further enhanced by adding enhancement nodes. To update the connection weights, the model utilizes ridge regression based on the pseudoinverse method. In comparison to deep learning approaches, BLS can be constructed with ease and expediency, even in the absence of high-performance computing resources. In addition, many advanced variants of BLS have emerged, such as the one proposed by Wang et al. (2019b), which improves its performance by utilizing the independence and flexibility of mapping features and enhancement nodes. In this article, we propose a novel broad attentive cross module based on BLS theory. This approach facilitates the efficient exploitation of higher-order feature interactions at the bit level.

Proposed model

In this section, we define the task associated with the model and provide a detailed explanation of its components.

Definition of the task

To facilitate analysis, the set of user attributes is denoted by AU, while the set of item attributes is denoted by AI. Attribute-value pairs are conceptualized as key-value pairs ( att, val), where att represents the name of the attribute and val signifies the corresponding value associated with that attribute. For instance, ( Age, 25) and ( Location, Beijing) indicate that the user’s Age is 25 and Location is Beijing, here Age∈AU and Location∈AU are considered as two instances of the user attribute. Let D be a dataset comprising N training data pairs, mathematically represented as D={(xi,yi)}1≤i≤N. For each training data pair (xi,yi)∈D, the term xi represents a data sample that consists of a set of user attribute-value pairs denoted as CiU, along with a set of item attribute-value pairs denoted as CiI. They are defined as follows:

(1) CU={CiU=(att,val)}att∈AU,i=1,⋯,nCI={CjI=(att,val)}att∈AI,j=1,⋯,mxn=CU∪CI.

It is important to note that the number of attribute-value pairs in a data sample is not fixed. This is because, in some cases, the data sample may contain missing attribute information or multiple instances of the same attribute within the sample, e.g., a book may span multiple disciplines.

Therefore, given an xn, which includes user and item attributes, as well as dataset D, this study aims to propose a recommendation model that facilitates the prediction of the maximum score. This task can be articulated as an optimization problem, which is represented in the following manner:

(2) arg⁡maxθ∗yn=F(θ,D,xn),

where yn∈R denotes implicit feedback, such as purchase and browsing, from the user on the item. F is the neural network model of recommendation, θ is the learning parameters of F.

Overview

As can be seen from Fig. 1, we have divided the model diagram of this article into a main framework diagram, Fig. 1A, and three detailed module diagrams: Fig. 1B self-attention module, Fig. 1C attribute fusion module, and Fig. 1D broad attention cross module. Where Fig. 1A further divides the GNN-A2 into the following three parts.

Figure 1 The overall architecture of GNN-A2.

Attribute embedding presentation. The user and item features in the data sample are constructed into the corresponding attribute graphs as well as adjacency matrices, respectively.

Attribute interaction subnetwork. This subnetwork comprises three components: the inner interaction module, the attribute fusion module, and the broad attentive cross module. The inner interaction module integrates GNN with self-attention mechanisms to exploit more beneficial feature interactions. The attribute fusion module combines user and item attribute representations into a cross attribute representation. The broad attentive cross module further enhances the refinement of higher-order feature interactions through a broad attention mechanism at the bit level.

Prediction layer. The final node representation is obtained by aggregating the node information, after which the prediction is obtained using the dot product.

Attribute embedding representation

We employ attribute graphs to represent users, items, and interactions between users and items, respectively. There are the user attribute graph GU, the item attribute graph GI, the attribute interaction graph GUI with user attributes as a center node, and the attribute interaction graph GIU with item attributes as a center node. In these graphs, attributes are depicted as nodes within the graph, connected edges between nodes to represent interactions between the attributes.

Firstly, we illustrate the embedding presentation of the user attribute inner interaction graph in terms of the construction of GU. In GU, user attributes are depicted as nodes within the graph. Each attribute att∈AU is embedded as a vector v∈Rd with dimensional size d. This process constructs a parameter matrix, which serves as an embedding index table. The various pair of attributes-values, denoted as (att,val), are represented by the vector eatt, which is computed using the following equation.

(3) eatt=val⋅vatt,

where vatt is the vector embedding for the same attribute att, consisting of all data samples. However, due to the varying potential values of val, they may exhibit different scalar multipliers on the vector. Initially, the vector eatt for each attribute att is assigned at random. For each data sample, the user attributes are denoted by the symbol eattU. The set of nodes in the user attribute graph is expressed as VU={eattU}att∈AU.

In GU, connected edges to represent the co-occurrence of attributes att1 and att2 are defined as their interaction, which is modeled by the function f(ea1,ea2):R2×d→Rl, where ea1 and ea2 represent the attribute-value pairs of att1 and att2 in the data samples, and l denotes the output dimension. The synergy of the interaction information in various data samples helps to reveal the interaction information between attributes that have not co-occurred. This enables the function f(⋅,⋅) to effectively learn attribute embeddings that reflect the synergistic relationship of attributes (Rendle, 2010). Such that, we can represent all possible edges through the utilization of f(⋅), thereby constituting a complete graph. The set of edges in the user attribute graph denoted as EU={f(ea1,ea2)}a1,a2∈AU. For ease of writing, we will directly replace atti with the subscript i in the narrative that follows. Thus, the user attribute graph is represented by GU=⟨VU,EU⟩. Similarly, the item attribute graph GI=⟨VI,EI⟩ can be obtained.

Secondly, we focus on the representation of GUI and GIU. Here, we utilize the adjacency matrix, denoted A, to represent the relationship between the user attributes nodes and the item attributes nodes. In the event of an interaction between user attribute ei and item attribute ej, the value of Aij is 1. Otherwise, it is 0. The adjacency matrix AUI∈A presents the attribute interactions that occur between users and items, with user attributes as the center node. And AIU∈A presents the attribute interactions that exist between items and users, with item attributes as the center node.

Attribute interaction subnetwork

Depending on the treatment, the inner interaction module, the cross interaction module, and the broad attentive cross module will be discussed separately in this subsection.

Inner interaction module

In GNN-A2, we use a GNN-based message-passing approach (Su et al., 2021b) combined with a self-attention mechanism to model the inner interaction of feature learning. The basic idea is that the squeeze-and-excitation (SE) block (Su et al., 2021a; Hu et al., 2017) is used to implement a self-attention mechanism before aggregation operation in GNN.

Figure 1B is a diagram of the self-attention module, which includes the compression, excitation, and scaling processes. Specifically, we first squeeze the feature graph by generating feature statistics using global average pooling, which facilitates the search for potential relationships between features. This process can be expressed as:

(4) ti=fsq(ei,ek)=1d(ei⊙ek),k=1,⋯,n,

where ti denotes the feature vector of node i after squeeze, ei is the embeding of node i, ⊙ denotes the element-wise product operator, d denotes the dimension size of embedding vector. Thus, we can obtain feature vectors of all attribute nodes, denoted as t={ti},i=1,⋯,n.

Subsequently, the feature values need to be synthesized. The high values obtained for the inner interaction do not necessarily imply that the two features should be considered similar. Given that inner interactions are employed to capture user (item) features, the process inherently exhibits complexity. Therefore, this synthesis method needs to fulfill two conditions: the ability to express nonlinear relations, and the ability to extract non-mutually exclusive relations. In this regard, we utilize a multilayer perceptron (MLP) with two full connection layers to calculate a weighting factor.

(5) α=fex(t,W)=σ(g(t,W))=σ(W2δ(W1t)),

where δ is the ReLU activation function, W1 and W2 are the learning weight matrixs, and σ is the sigmoid function.

Then the weighting factor α and the squeezed feature value ti are merged to recalibrate the feature graph. The specific merging method is represented as follows:

(6) xi^=fscale(α,ti)=αti⊕ti,

where ⊕ is the element-wise addition operator.

Finally, we employ element-wise summation to aggregate the results of modeling inner interactions among each node into new nodes. This aggregation can be expressed by

(7) qi=∑j∈N(i)x^j,

where qi represents the results of message passing of node i, N(i) represents the neighbor nodes set of node i.

Cross interaction module

Figure 1C shows the attribute fusion structure, which integrates an attention mechanism to evaluate the correlations among attributes that constitute the cross interaction module. The purpose of cross interaction is to combine the attribute representations of users and items into a cross attribute representation. Traditional approaches typically model different classes of data separately, based on the type of attributes. Although subsequent advancements have been made, they continue to overlook the significance of the different cross interaction attributes that exist between user attributes and item attributes. In this approach, we adopt a dual strategy. On one hand, we choose to disregard attribute types and treat all attribute values uniformly as a single class. This method enables us to capture complex relationships across different classes while simplifying modeling complexity, thereby facilitating the propagation of embeddings. On the other hand, we implement relevance computation that incorporates an attention mechanism to effectively differentiate the varying impacts of diverse cross attributes on feature learning.

By disregarding the distinction between various types of attributes, we can treat different attribute categories in a uniform manner. For instance, both user attributes (which are multi-valued) and item attributes (also multi-valued) can be represented as a set of multi-valued relationships within a unified “one-to-many” framework. This facilitates the alignment of different types of attributes and captures the relationships through a unified similarity calculation method.

Taking cross attribute graph of user GUI as a case study, we assess the impact of an item on a user by calculating the correlation between user attributes and item attributes. Firstly, we calculate the correlation between item attributes and user attributes by using cosine similarity.

(8) si,jUI=cosine_sim(eiU,ejI)=(eiU)T⋅ejI||eiU||2||ejI||2,j=(1,2,⋯,m=|N(i)|),

where eiU and ejI denote the user attribute node and item attribute node, respectively. Here, ejI is the jth neighbor node of node eiU in the graph GUI, N(i) is the neighbor nodes set of node i in GUI. The correlation score si,jUI is positively correlated with influence, indicating a greater degree of preference as the score increases. For instance, if female users exhibit a preference for skin care products, the relevance score of the node pair <female, skin care> should be high.

Next, the softmax function is employed to normalize the correlation sUI, thereby deriving the significance of item attributes with user attributes.

(9) γi,jUI=exp⁡(si,jUI)∑k=1mexp(si,kUI).

Subsequently, the cross attribute representation of the user and item is given by Eq. (10):

(10) ziU=∑j=1m((γi,jUIejI)⊙eiU),

where ⊕ is the element-wise product operator, ∑ is the element-wise addition operator.

Similarly, the cross attribute representation of each item in GIU can be obtained zI. This concludes the cross interaction between user and item attributes.

Broad attentive cross module

The attribute interaction module discussed in the previous section effectively captures vector-level feature interactions and integrates all first-order interaction information. However, it does not fully leverage higher-order interaction information. To address this limitation, our model incorporates a broad attentive cross module following the attribute interaction module, which is specifically designed to enhance higher-order feature interactions at the bit level. This module dynamically adjusts the weights of important nodes, while reducing the weights of less significant nodes.

In the traditional broad learning system (BLS) (Chen & Liu, 2018), feature nodes are generated independently from one another through feature mapping, without considering the interrelationships among feature nodes. This approach overlooks the intrinsic relevance of the input data. In this study, we contend that effective modeling of feature interactions should be capable of capturing higher-order cross features with complex intrinsic correlations. Therefore, we draw inspiration from time-delayed neural networks (Chen, Liu & Feng, 2019; Lipton, 2015; Chaturvedi et al., 2016) to implement a cascading recursive reconstruction of the BLS in the form of recurrent, thereby enhancing the correlation between nodes.

As illustrated in Fig. 1D, the broad attentive cross module incorporates mapping features and enhancement nodes, enhancing the interconnections between nodes through full connectivity to optimize feature mapping performance. Specifically, the detailed procedures of broad attentive cross module are as follows.

Step 1: We construct cascade mapping features through a two-layer multilayer perceptron (MLP), which can be articulated as follows:

(11) Fr=ϕ(Wh2ϕ(Wh1Fr−1+Bh1)+Bh2),r=1,⋯,R,

where Wh1, Wh2, Bh1, and Bh2 are learning weights and biases, respectively. R is the total number of mapping features. ϕ represents the LeakyReLu activation function. It should be noted that F0={qi⊕zi}i=1n. Here, ⊕ signifies elementwise addition. Thus, the final mapping feature nodes can be obtained by concatenating all Fr, denoted as F=Δ[F1,…,FR].

Step 2: We utilize the same MLP block to perform enhanced feature mapping. The processing can be defined as:

(12) Ps=ϕ(Wg2ϕ(Wg1Ps−1+Bg1)+Bg2),s=1,⋯,S,

where Wg1, Wg2, Bg1, and Bg2 are learning parameters. ϕ is the LeakyReLu activation function. All enhanced features are concatenated to create the final enhanced feature nodes, denoted as P=Δ[P1,…,PS].

Step 3: After acquiring the mapping nodes F and the enhancement nodes P, applying F and P allows us to ascertain the broad attentional weights ξ of the global-aware nodes. This parameter aids in evaluating the significance of each higher-order feature interaction.

(13) ξ=[F||P]WξWξ=[F||P]+ξ=Δ+ξΔ+=limλ→0⁡(λI+ΔTΔ)−1ΔT,

where Wξ represents the matrix of trainable connection weights, || is the concatenation operation, Δ+ is the pseudoinverse of Δ=[F||P]. Here, λ is an adjustment parameter, initially set to a value of 0.1, and I represents the identity matrix.

Step 4: Ultimately, the representation of higher-order cross features is as follows:

(14) Γi=(W2b(qi⊗zi))⊙(W1b(qi⊕zi))⊙σ(pooling(ξ)),

where W1b and W2b represent two matrices of trainable network weights. The symbols ⊙, ⊗, and ⊕ denote the Hadamard product, the inner product, and the element-wise addition, respectively. Furthermore, ξ represents the broad attentive weights of global-aware nodes, while σ signifies applying a sigmoid activation function.

In this module, interactions among higher-order features occur at the bit level. Broad attention weights indicate the significance of these higher-order feature interactions, allowing higher-order features to achieve a finer-grained representation in each dimension.

Prediction layer

After acquiring the higher-order interaction information, we integrate the initial node representation ei with it and utilize the node fusion function fGRU∈R2×d→Rd to obtain the representation of the fused node, expressed by:

(15) e^i=fGRU(ei⊕Γi),

where e^i denotes the fused node representation of node ei, and ⊕ denotes the concatenate operator. After transforming the fused node representations into a new graph structure, we employ element-wise addition to aggregate the graph into vector forms. The final representations of the user attribute graph and the item attribute graph are as follows:

(16) vGU=∑i=1ne^iU,vGI=∑j=1me^jI.

Finally, we use the dot product to match the two vectors and compute the final prediction score:

(17) y=vGUvGI.

Loss function

In this article, we employ the L2 regularization criterion for all parameters of the GNN-A2, and the corresponding loss function is expressed as follows::

(18) L(θ)=1N∑n=1Nynlog⁡yn+(1−yn)log⁡(1−yn)+λ(||θ||2),

where θ is the learning parameters of the GNN-A2 model, λ is the regularization parameter term.

Experiments and results

In this section, we conduct a comprehensive series of experiments to assess the performance of GNN-A2 on real-world datasets. This evaluation aims to address the following questions: RQ1: How does our model’s performance in terms of accuracy for preference recommendations compare to other baseline models that consider user attributes and item attributes?

RQ2: How does each module in GNN-A2 affect providing accurate predictions?

RQ3: How do differences in parameter settings affect the performance of our model?

Experimental settings

Datasets

To assess the performance of the GNN-A2, we conducted a series of experiments utilizing three publicly available baseline datasets: MovieLens 1M (http://grouplens.org/datasets/movielens/1m/), Book-crossing (http://www2.informatik.uni-freiburg.de/cziegler/BX/), and Taobao (https://tianchi.aliyun.com/dataset/56). The statistics summaries of the datasets are presented in Table 1. Movielens 1M (Harper & Konstan, 2015) is a broadly used recommender system dataset that contains user ratings for movies, we categorize ratings exceeding 3 as indicative of a positive evaluation. Each entry in this dataset contains a user and a movie, encompassing several basic attributes. To make this dataset richer, additional information about the movie is added to it, such as the director and actors. Book-crossing (Ziegler et al., 2005) is a dataset that encompasses both implicit and explicit ratings of books provided by users. Each entry in this dataset contains a user and a book, encompassing several basic attributes. Taobao (Zhou et al., 2018) comprises a compilation of advertisement click logs sourced from the Taobao website. These logs record the clicking behavior of users, and each log entry contains information about users with specific attributes, e.g., gender and age, as well as information about the advertisements they see, e.g., the category and the price.

Table 1 Dataset statistics.

Dataset	#Data	#User	#Item	#User attr.	#Item attr.	
MovieLens 1M	1,149,238	5,950	3,514	30	6,944	
Book-crossing	1,050,834	4,873	53,168	87	43,157	
Taobao	2,599,463	4,532	371,760	36	434,254	

For the MovieLens 1M and Book-crossing datasets, which contain explicit ratings, we convert these ratings into implicit feedback. In MovieLens 1M, ratings above 3 are considered positive, while for Book-crossing, all ratings are treated as positive due to its sparsity. We then randomly select an equal number of negative samples to match the number of positive samples for each user. To maintain dataset quality, we filter out users who have fewer than 10 positive ratings in MovieLens 1M, and fewer than 20 positive ratings in Book-crossing and Taobao.

Evaluation metrics

We employ several different metrics to assess the performance of both our model and the baseline model. The area under the curve (AUC) reflects the model’s ability to distinguish between different categories. Logloss is a metric that evaluates the predictive accuracy of a binary classification model, which measures the difference between the predicted probabilities generated by the model and the actual outcomes. The normalized discounted cumulative gain (NDCG) is a commonly used metric for evaluating the performance of recommender systems, which considers the rank order of items in the recommendation list. In particular, we used NDCG@k to assess the top k recommended items of the model, where for k we chose 5 and 10 as evaluation points.

Baselines

To validate performance advantages, we conduct a comparison between the GNN-A2 and several baseline models as follows.

FM (Rendle, 2010) captures the interactions between different features through the dot product of feature vectors, subsequently integrating these interaction effects to form the final prediction.

NFM (He & Chua, 2017) focuses on effectively capturing second-order feature interactions by feeding the elemental product of feature vectors as input to a dual interaction pooling layer.

Wide & Deep Learning (W&D) (Cheng et al., 2016) effectively memorizes sparse feature interactions by combining linear modeling and deep learning techniques, demonstrating the ability to scale to previously unseen interactions.

DeepFM (Guo et al., 2017) further combines FM and MLP to enable a more comprehensive interaction analysis. This is achieved by connecting the embedding vectors of all attributes and learning further interactions through MLP.

AutoInt (Song et al., 2019) explicitly models the interactions between features by employing the multi-head self-attention mechanism with residual networks.

Fi-GNN (Li et al., 2019) employs a feature graph to represent data samples. In this framework, each node in the graph corresponds to a distinct feature domain, and interactions between nodes are modeled through a multi-head self-attention mechanism.

L0-norm Statistical Interaction Graph neural Network (L0-SIGN) (Su et al., 2021a) improves the efficiency of recommender systems by treating each data sample as a graph, detecting the most beneficial feature interactions in it, and utilizing only these interactions to make recommendations.

GMCF (Su et al., 2021b) constructs a graph matching structure by identifying and aggregating two different types of attribute interactions, which are efficiently captured to enhance recommendations.

Co-Action Network (CAN) (Bian et al., 2020) uses the MLP structure to explore the connection between the behavior patterns of the past users and their interest in specific items, approximating the empirical features interactions.

All of these approaches work to enhance the performance and accuracy of recommender systems through different technical means.

Experimental setup

For dataset division, we adopted a random assignment method to divide the dataset into three parts: a training set, a validation set, and a test set, maintaining a 6:2:2 ratio. In this case, the validation set serves to optimize the model parameters, whereas the test is employed to assess the final performance of the model. In the experiment, the node representation dimension was set to 64. The MLP consists of a single hidden layer, with the number of neurons in each layer being four times the input dimension. The learning rate was set to 1×10−3. The regularization parameter λ is 1×10−5. We used binary cross-entropy to choose the loss function. We used the Adam optimizer (Kingma & Ba, 2014), which dynamically adjusts the learning rate of each parameter according to the first and second moments of the gradient, thus ensuring more efficient training.

Experimental results and analysis

Performance comparison

We compare the proposed GNN-A2 with nine competing baselines on MovieLens 1M, Book-crossing, and Taobao datasets. The experimental results are presented in Table 2. To enhance the clarity of the experimental results, the optimal results for each dataset are highlighted in bold, and the suboptimal ones are indicated with underlining. The following conclusions can be drawn:

Table 2 Performance comparison between GNN-A2 and baselines.

To enhance the clarity of the experimental results, the optimal results for each dataset are highlighted in bold, and the suboptimal ones are indicated with underlining.

Model	MovieLens 1M	Book-crossing	Taobao	
	AUC	Logloss	NDCG@5	NDCG@10	AUC	Logloss	NDCG@5	NDCG@10	AUC	Logloss	NDCG@5	NDCG@10	
FM	0.8761	0.4409	0.8143	0.8431	0.7417	0.5771	0.7616	0.8029	0.6171	0.2375	0.812	0.1120	
NFM	0.8985	0.3996	0.8486	0.8832	0.7988	0.5432	0.7989	0.8326	0.6550	0.2122	0.0997	0.1251	
W&D	0.9043	0.3878	0.8538	0.8869	0.8105	0.5366	0.8048	0.8381	0.6531	0.2124	0.0959	0.1242	
Deep-FM	0.9049	0.3856	0.8510	0.8848	0.8127	0.5379	0.8088	0.8400	0.6550	0.2115	0.0974	0.1243	
AutoInt	0.9034	0.3883	0.8619	0.8931	0.8130	0.5355	0.8127	0.8472	0.6434	0.2146	0.0924	0.1206	
Fi-GNN	0.9063	0.3871	0.8705	0.9029	0.8136	0.5338	0.8094	0.8522	0.6462	0.2131	0.0986	0.1241	
L0-SIGN	0.9072	0.3846	0.8849	0.9094	0.8163	0.5274	0.8148	0.8629	0.6547	0.2124	0.1006	0.1259	
GMCF	0.9127	0.3789	0.9374	0.9436	0.8228	0.5233	0.8671	0.8951	0.6679	0.1960	0.1112	0.1467	
CAN	0.9133	0.3773	0.9396	0.9442	0.8235	0.5143	0.8722	0.8996	0.6776	0.1919	0.1130	0.1494	
GNN-A2	0.9101	0.3846	0.9511	0.9506	0.8400	0.4956	0.9003	0.9137	0.6715	0.1944	0.1159	0.1526	
Improv	–	–	1.22%	0.68%	2.00%	3.64%	3.22%	1.57%	–	–	2.57%	2.14%	

GNN-A2 outperforms most of the baseline models on various metrics. Notably, GNN-A2 achieves the optimal performance on NDCG@10 over three datasets, with values of 0.9506, 0.9137, and 0.1526, corresponding to respective improvements of 0.68%, 1.57%, and 2.14% compared to the SOTA models. However, its performance is less impressive in the MovieLens 1M dataset and the Taobao dataset, which have fewer user attribute features. This is because more attribute features usually capture a wider range of cross features, leading to better results, especially in large, sparse datasets.

FM and AFM lack deep learning mechanisms, relying only on dot products for feature interactions, which limits their performance. To model complex interactions more effectively, advanced techniques like MLP are needed. Our model leverages this approach to better capture interactions in user and item attribute graphs, enabling more accurate predictions.

GNN-based models like Fi-GNN, GMCF, and L0-SIGN outperform other algorithms, demonstrating the advantage of GNN in capturing complex attribute interactions. GNN-A2 enhances this by incorporating an attention mechanism, leading to better performance than L0-SIGN and GMCF.

In analyzing the three different evaluation datasets, we observe that most of the higher-order models exhibit more significant learning compared to the learning of lower-order feature interactions. This phenomenon is consistent with our understanding that higher-order feature interactions have a greater impact on prediction accuracy.

These findings validate the ability of the GNN-A2 model to parse structured features of user and item attributes, which in turn enables accurate prediction results.

Ablation study

To investigate the impact of the self-attention mechanism, attribute fusion, and higher-order interactions on model prediction, we conducted ablation experiments by systematically removing each component: the self-attention mechanism, the attribute fusion module, and the broad attentive cross module. The self-attention mechanism is denoted as Attention, the attribute fusion module is denoted as Fusion, and the broad attention cross module is denoted as Broad. We set up the following three GNN-A2 variants: GNN-A2-Attention removing the self-attention mechanism, GNN-A2-Fusion removing the attribute fusion module, and GNN-A2-Broad removing the broad attentive cross module. If both modules are available, our model is established. This approach allowed us to assess their contributions to GNN-A2’s performance. To ensure fair comparisons between the models, we standardized the parameters in all variant models. The experimental results are presented in Table 3. The trends observed in Logloss and NDCG@5 were found to be similar to those of AUC and NDCG@10. Therefore, they have been omitted from the analysis. Ultimately, our analysis of the experimental results revealed that the self-attention mechanism, the attribute fusion module, and the broad attentive cross module all contributed to enhancing the performance of the model to varying degrees. This finding substantiates the feasibility of these modules.

Table 3 Ablation study.

To enhance the clarity of the experimental results, the optimal results for each dataset are highlighted in bold, and the suboptimal ones are indicated with underlining.

Model	MovieLens 1M	Book-crossing	Taobao	
	AUC	NDCG@10	AUC	NDCG@10	AUC	NDCG@10	
GNN-A2-Attention	0.9053	0.9457	0.8253	0.9108	0.6682	0.1501	
GNN-A2-Fusion	0.9072	0.9415	0.8307	0.9061	0.6693	0.1497	
GNN-A2-Broad	0.9087	0.9403	0.8341	0.9032	0.6706	0.1485	
GNN-A2	0.9101	0.9506	0.8400	0.9137	0.6715	0.1526	

Fusion algorithm research

GNN-A2 obtains fused node representations by summarizing raw node information and higher-order interaction information. In addition to Gated Recurrent Unit (GRU), we further evaluated the effectiveness of element summing and MLP as fusion node strategies.

Table 4 shows the experimental results for three fusion algorithms. The results indicate that GRU outperforms others in aggregating original node and higher-order interaction information into fused node representations. In contrast, the summation (SUM) method performed worst in most cases, highlighting the complexity of merging both types of information and the need for more powerful algorithms.

Table 4 Effects of different fusing algorithms on model performance.

To enhance the clarity of the experimental results, the optimal results for each dataset are highlighted in bold, and the suboptimal ones are indicated with underlining.

Fusing algorithms	MovieLens 1M	Book-crossing	Taobao	
	AUC	NDCG@10	AUC	NDCG@10	AUC	NDCG@10	
SUM	0.9049	0.9431	0.8295	0.8935	0.6606	0.1459	
MLP	0.9031	0.9455	0.8267	0.9042	0.6657	0.1492	
GRU	0.9101	0.9506	0.8400	0.9137	0.6715	0.1526	

Hyperparameters study

In this subsection, we assess GNN-A2 under various hyperparameter configurations. Specifically, we examine the dimensionality of node representations and the depth of the MLP utilized in the model.

Figure 2 illustrates the performance comparison of our model under different node representation dimensions (D). Analyzing the graphs, we can observe that GNN-A2 achieves optimal performance for most of the datasets when the dimension is set to 64. It should be noted that the prediction is better in 256 dimensions for the Book-Crossing dataset. This insight demonstrates that an increase in the dimensionality of node representations does not necessarily correlate with a proportional enhancement in prediction accuracy. The reason is that as the dimensionality of node representation increases, the model parameters that need to be learned also increase. This increase can potentially heighten the risk of overfitting within the model.

Figure 2 The performance of GNN-A2’s with different attribute embedding dimensions.

Next, we analyze how varying the number of hidden layers in the MLP impacts our model’s performance. Specifically, our model employs an MLP with a single hidden layer to capture inner interactions during message passing. To evaluate the effect of different layer counts, we test the model using 0, 1, 2, and 3 hidden layers, denoted as GNN-A2-0, GNN-A2-1, GNN-A2-2, and GNN-A2-3, respectively. Here, GNN-A2-0 indicates that the MLP solely applies a linear transformation from node representations to interaction modeling outputs. Each hidden layer in these configurations is set to have the same number of units, specifically 4D.

Table 5 presents the performance results for GNN-A2 with varying numbers of MLP layers. It is evident that GNN-A2-0, which uses an MLP without hidden layers, performs significantly worse than other configurations. This highlights the advantage of employing a non-linear approach to better extract meaningful information from inner interactions, compared to a simple linear transformation. The best performance is achieved with a single hidden layer in the MLP. This suggests that increasing the number of hidden layers does not necessarily enhance performance, as deeper MLPs may lead to overfitting. Thus, a single hidden layer is sufficient for effectively analyzing inner interactions in our models.

Table 5 The impact of varying MLP depths on GNN-A2’s performance.

To enhance the clarity of the experimental results, the optimal results for each dataset are highlighted in bold, and the suboptimal ones are indicated with underlining.

Model	MovieLens 1M	Book-crossing	Taobao	
	AUC	NDCG@10	AUC	NDCG@10	AUC	NDCG@10	
GNN-A2-0	0.9050	0.9421	0.8296	0.9015	0.6702	0.1467	
GNN-A2-1	0.9101	0.9506	0.8400	0.9137	0.6715	0.1526	
GNN-A2-2	0.9091	0.9489	0.8387	0.9090	0.6709	0.1503	
GNN-A2-3	0.9076	0.9456	0.8367	0.9073	0.6704	0.1489	

Conclusions

In this study, we propose a GNN-A2 aware-attribute recommendation model that incorporates a self-attention mechanism, attribute fusion, and broad attention. We employ a self-attention mechanism to assign varying weights to inner interactions between attributes. We use attribute fusion to effectively capture the cross interaction between user and item attributes. We introduce the broad attentive cross module to capture higher-order interaction information, which improves the model’s prediction accuracy and generalization ability. Experimental results indicate that GNN-A2 is effective, mostly outperforming the baseline model on three standard datasets widely used in recommender systems.

However, despite the promising results, there will still be limitations in its scalability when applied to very large datasets, particularly in real-world scenarios where data sparsity and volume are substantial. As the number of users and items increases, the computational complexity of GNN-based models can become prohibitive. Although techniques such as mini-batch and distributed graph processing can partially alleviate this issue, effectively handling very large-scale graphs while maintaining model performance remains a significant challenge. In the future, we will concentrate on exploring efficient methods to extend GNN-A2 to larger datasets, such as hierarchical graph-based representations and parallelized message-passing schemes. Concurrently, we are also interested in developing techniques to adapt GNN-A2 for real-time updates. This would involve incorporating techniques for incremental learning or online training to continuously refine the model based on new user behavior.

Supplemental Information

Supplemental Information 1 Contains scoring files from all three datasets.

Additional Information and Declarations

Competing Interests

The authors declare that they have no competing interests.

Author Contributions

MingXue Liu conceived and designed the experiments, performed the experiments, analyzed the data, performed the computation work, prepared figures and/or tables, authored or reviewed drafts of the article, and approved the final draft.

Min Wang conceived and designed the experiments, analyzed the data, authored or reviewed drafts of the article, and approved the final draft.

Baolei Li conceived and designed the experiments, authored or reviewed drafts of the article, and approved the final draft.

Qi Zhong analyzed the data, prepared figures and/or tables, and approved the final draft.

Data Availability

The following information was supplied regarding data availability:

The Movielens 1m dataset is available at: https://grouplens.org/datasets/movielens/1m.

The Taobao dataset is available at: https://tianchi.aliyun.com/dataset/dataDetail?dataId=56.

The datasets and code are available at GitHub: https://github.com/LMXue7/GNN-A2.

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
