# Peer review of "Collaborative filtering based on GNN with attribute fusion and broad attention"

_PeerJ Computer Science, doi:10.7717/peerj-cs.2706_

## Round 0.1 · original submission · Minor Revisions

Dear Authors,
Your paper has been revised. Based on the reviewers' evaluation, minor revisions are needed before it is considered for publication in PEERJ Computer Science. More precisely, the following points must be faced in the revised version of your paper:
1)The related work section could be further strengthened by discussing the latest state-of-the-art graph collaborative filtering models that leverage self-supervised learning techniques.
2) The abstract needs to be improved, indicating specific quantitative results or improvement percentages compared to baseline models.
3) An explanation of how the attribute fusion mechanism works or what makes it novel compared to existing fusion approaches must be added. Furthermore, the "broad attentive cross module" description must be improved, specifying how the proposed approach captures higher-order interactions differently from existing methods.

Reviewer 1 ·

Basic reporting

This paper proposes a collaborative filtering recommendation model called GNN-A2 that uses graph neural networks to improve performance. It addresses two key issues in existing models - the indiscriminate aggregation of all attributes, and the lack of exploiting higher-order interaction information.

Experimental design

The authors compared the performance of GNN-A2 against several baseline models, including traditional recommendation approaches as well as more recent deep learning-based methods.

Validity of the findings

The statistical analyses and modeling approaches used appear to be robust and sound. The authors take steps to control for confounding factors and ensure the validity of their findings. Overall, the paper presents a strong case for the validity and reliability of the findings from the GNN-A2 model.

Additional comments

The attribute fusion strategy for modeling cross-interactions requires more detailed explanation. While the paper states that this module assesses the correlation between user attributes and item attributes using cosine similarity to assign weights, more specifics on the mathematical formulation and implementation would be helpful for readers to fully understand this key component.

The broad attentive cross module, which is designed to fuse information from the inner interactions and cross-interactions, could benefit from a more thorough walkthrough. The paper mentions that this module "dynamically evaluates the degree to which different feature interactions contribute to the final prediction", but a clearer step-by-step description of how this module operates would strengthen the presentation of this novel aspect of the GNN-A2 framework.

The related work section could be further strengthened by including discussion of the latest state-of-the-art graph collaborative filtering models that leverage self-supervised learning techniques. For example, the authors could consider incorporating discussion of papers such as "Candidate-aware graph contrastive learning for recommendation" (SIGIR 2023) and "Self-supervised hypergraph transformer for recommender systems" (KDD 2022). These recent works have demonstrated impressive performance in graph-based recommendation tasks by employing novel self-supervised learning approaches.

Cite this review as

Reviewer 2 ·

Basic reporting

1. The article demonstrates a professional level of English, with a clear and precise style appropriate for an academic manuscript. The language used is formal and technically accurate, suitable for a scientific audience. However, some sentences in the introduction and results sections could benefit from more concise wording to enhance readability.
2. The article provides a solid review of related work, offering comprehensive citations to establish the context of graph neural networks (GNNs) and collaborative filtering (CF). It effectively discusses prior methodologies, including matrix decomposition, factorization machines, and GNN-based approaches. The authors also position their contribution relative to existing models, such as GMCF and Fi-GNN. The literature coverage is sufficient and provides a clear rationale for the proposed model.
3. The structure of the manuscript is professional and adheres to academic standards. Sections such as the introduction, related work, proposed model, experiments, and conclusions are well-defined. Figures and tables are appropriately labeled, with clear legends and descriptions. For example:
While the figures are informative, some could be simplified for better visual clarity, particularly for complex diagrams like the overall framework of GNN-A2 (Figure 1.)!
4. The article references three publicly available datasets (MovieLens 1M, Book-crossing, and Taobao) and provides a table summarizing their statistics. However, it does not explicitly share (how it is used) or link to the raw data used for experiments within the manuscript. Although the datasets are widely recognized, including a direct link or supplementary material to facilitate reproducibility would enhance transparency (URL, Link, reference, citation, source, web address).
5. The manuscript is largely self-contained, presenting a clear hypothesis and aligning the experimental results with the research objectives. The authors describe the GNN-A2 model and evaluate its performance using appropriate metrics such as AUC, NDCG@5, and Logloss. Ablation studies and comparative analyses with baseline models further validate the hypotheses. However, the description of how the model parameters were optimized could be elaborated for greater clarity. Also NDCG@20 can be shared.
6. The article does not include formal theorems or proofs as it primarily focuses on algorithmic and experimental contributions. However, all terms, such as "attribute fusion" and "broad attention module," are clearly defined within the context of the proposed model. Mathematical formulations for model components and loss functions are well-explained and adequately supported by equations.
Briefly in this section:
The manuscript demonstrates an original scientific contribution to the field by proposing the GNN-A2 model, which addresses limitations in prior collaborative filtering methods. It aligns well with academic standards in terms of structure, content, and clarity.

Experimental design

1. The manuscript presents original primary research that fits well within the field of computer science, particularly focusing on recommender systems and graph neural networks (GNNs). The use of the GNN-A2 model, which incorporates attribute fusion and broad attention mechanisms, is a novel approach to addressing limitations in existing collaborative filtering (CF) methods.
2. The research question is clear and centers on addressing two main issues in existing GNN-based CF models: indiscriminate aggregation of attributes and the underutilization of higher-order interactions. The question is relevant to the field as it seeks to improve recommendation system accuracy and generalization—key challenges in practical applications such as e-commerce, media streaming, and social platforms. The question is meaningful because it directly targets practical and theoretical gaps in the literature.
3. The authors explicitly state the limitations of existing models, such as GMCF and Fi-GNN, in distinguishing between inner and cross interactions and in capturing higher-order attribute interactions. They argue that their proposed GNN-A2 model overcomes these limitations through its novel attribute fusion strategy and broad attentive cross module. The paper effectively communicates its contribution to bridging this gap by demonstrating superior performance across multiple datasets.
4. The investigation appears rigorous, with extensive experiments conducted on three benchmark datasets (MovieLens 1M, Book-crossing, and Taobao). The paper evaluates the model using standard metrics (AUC, NDCG@k, and Logloss) and performs ablation studies to validate the contributions of individual components (e.g., self-attention, attribute fusion, and broad attentive modules). The inclusion of baseline comparisons against nine competing models ensures a robust analysis.
From an ethical perspective, the use of publicly available datasets mitigates concerns about privacy or data misuse.
5. The methods section provides substantial details about the architecture of GNN-A2, including:
-Mathematical definitions of inner and cross interactions.
-Descriptions of the self-attention mechanism, attribute fusion, and broad attentive cross modules.
-The loss function and experimental setup, including hyperparameters and optimization strategies.
While the description is thorough, the exact preprocessing steps for the datasets and code availability are not mentioned. Including these details would improve replicability.
Briefly in this section:
The research is well-aligned with the journal’s scope, addressing a meaningful research question and filling a knowledge gap. The investigation is conducted rigorously and is supported by detailed methodological explanations. Minor improvements in replicability could further enhance the work’s impact.

Validity of the findings

1. The manuscript demonstrates novelty by addressing critical limitations of existing GNN-based collaborative filtering models, specifically the challenges of indiscriminate attribute aggregation and the underutilization of higher-order interactions. The GNN-A2 model, with its attribute fusion and broad attention mechanisms, represents a meaningful extension to the current state-of-the-art in recommender systems.
The authors provide a strong rationale for their approach by comparing their model’s performance against baseline models and articulating how GNN-A2’s features contribute to the field. The results highlight measurable improvements in metrics like AUC and NDCG@k, reinforcing the model's practical impact. While replication is encouraged through the use of publicly available datasets and detailed model descriptions, explicitly sharing implementation code (https://github.com/LMXue7/GNN-A2) would further facilitate reproducibility and benefit the research community.
2. The authors utilize three well-known benchmark datasets: MovieLens 1M, Book-crossing, and Taobao, ensuring that the data is robust and representative of real-world recommendation scenarios. The datasets are statistically diverse, encompassing user ratings, behavioral logs, and attribute information, which helps validate the model's generalizability.
The results are presented with statistical rigor, comparing GNN-A2 to nine baseline models using established metrics. However, while the datasets are publicly accessible, the manuscript does not explicitly include links or details about preprocessing steps, which limits transparency. Sharing processed data or detailed preprocessing scripts would strengthen the robustness of the results and support controlled replication.
3. The conclusions are clear and directly tied to the research question, emphasizing the contributions of the GNN-A2 model in improving recommendation accuracy and generalization. The authors effectively discuss how each model component (e.g., self-attention, attribute fusion, broad attention) contributes to the observed performance improvements.
The experimental results are directly used to support the conclusions, with no overstatements or extrapolations beyond the presented data. However, a more detailed discussion of potential limitations (e.g., scalability for extremely large datasets or interpretability concerns in real-world deployments) would provide a balanced perspective.
Briefly in this section:
The manuscript effectively assesses the impact and novelty of the GNN-A2 model, providing robust data and well-stated conclusions linked to the research question. Minor improvements in data transparency and a discussion of limitations would enhance the manuscript’s contribution to the literature.

Additional comments

Recommendations for Improvement:
1. The abstract lacks specific quantitative results or improvement percentages compared to baseline models, making it difficult to assess the magnitude of improvement.
2. There's no clear explanation of how the attribute fusion mechanism works or what makes it novel compared to existing fusion approaches.
3. The description of the "broad attentive cross module" is vague and doesn't specify how it captures higher-order interactions differently from existing methods. While the high-level architecture is described, the paper lacks detailed mathematical formulations of the proposed modules (especially the broad attentive cross module). This makes it difficult to fully understand the technical contributions and reproduce the results. What is the specific formulation of the broad attention mechanism?
4. Citing and explaining a, b, c, and d in Figure 1 within the paragraph will make it easier to read and follow.
5. In the Conclusion section, researchers need to share a plan for their next studies or share plans that will serve as a basis for other researchers. In other words, what are the next studies or future works?
6. Discuss potential strategies to reduce the model's complexity and improve its efficiency for real-time applications.
7. Hyperparameter optimization experiment and results can be shared.
8. I recommend that you read these four articles. You can add them to your study.
>>Zhu, X.; Zhang, Y.; Wang, J.; Wang, G. Graph-enhanced and collaborative attention networks for session-based recommendation. Knowl.-Based Syst. 2024, 289, 111509.
>>Pan, Z.; Cai, F.; Chen, W.; Chen, C.; Chen, H. Collaborative Graph Learning for Session-based Recommendation. ACM Trans. Inf. Syst. 2022, 40, 1–26.
>>Pan, Z.; Cai, F.; Ling, Y.; de Rijke, M. An Intent-guided Collaborative Machine for Session-based Recommendation. In Proceedings of the 43rd International ACM SIGIR Conference on Research and Development in Information Retrieval, SIGIR ’20, Xi’an, China, 25–30 July 2020; pp. 1833–1836.
>>Luo, A.; Zhao, P.; Liu, Y.; Zhuang, F.; Wang, D.; Xu, J.; Fang, J.; Sheng, V.S. Collaborative Self-Attention Network for Session-based Recommendation. In Proceedings of the Twenty-Ninth International Joint Conference on Artificial Intelligence (IJCAI-20), Yokohama, Japan, 11–17 July 2020; pp. 2591–2597.

Cite this review as

---

## Round 0.2 · accepted · Accept

Dear Authors,

Yor paper has been revised. It has been accepted for publication in PEERJ Computer Science. Thank you for your fine contribution.

Reviewer 1 ·

Basic reporting

The proposed GNN-A2 model introduces several key innovations, including a self-attention mechanism for inner interactions, an attribute fusion strategy for cross interactions, and a broad attentive cross module to extract higher-order interactions. These advancements enable GNN-A2 to outperform state-of-the-art baselines on multiple recommendation benchmarks.

Experimental design

The authors investigated the impact of the self-attention mechanism, attribute fusion module, and broad attentive cross module by systematically removing each component. The results demonstrate the contributions of these components to the overall performance of GNN-A2.

Validity of the findings

The validity of the findings in this work is supported by the experimental results presented. The authors conducted a thorough performance comparison against a diverse set of state-of-the-art baseline models on three widely used benchmark datasets.

Additional comments

N/A

Cite this review as